# Slight Water Loss Combined with Methyl Jasmonate Treatment Improves *Actinidia arguta* Resistance to Gray Mold by Modulating Reactive Oxygen Species and Phenylpropanoid Metabolism

**DOI:** 10.3390/foods14244311

**Published:** 2025-12-14

**Authors:** Xinqi Liu, Qingxuan Wang, Feiyang Wang, Baodong Wei, Qian Zhou, Shunchang Cheng, Yang Sun

**Affiliations:** 1College of Food Science, Shenyang Agricultural University, Shenyang 110866, China; 13048234832@163.com (X.L.); 15922066633@163.com (Q.W.); wfy9069@163.com (F.W.); bdweisyau@163.com (B.W.); 66zhouqian@syau.edu.cn (Q.Z.); 2Liaoning Economic Forest Research Institute, Dalian 116031, China

**Keywords:** *Actinidia arguta*, *Botrytis cinerea*, postharvest water stress, methyl jasmonate, phenylpropanoid metabolism, reactive oxygen species, induced resistance

## Abstract

In this study, we aimed to elucidate the mechanism through which treatment with slight water loss combined with methyl jasmonate (MeJA) regulates gray mold development in *Actinidia arguta*, focusing on reactive oxygen species (ROS) and phenylpropanoid metabolism. The results showed that water loss alone, MeJA alone, and their combination each reduced the incidence of disease, with the combined treatment showing the greatest efficacy. At the end of the storage period, the combined treatment enhanced the activities of superoxide dismutase (SOD), polyphenol oxidase (PPO), peroxidase (POD), phenylalanine ammonia-lyase (PAL), cinnamate 4-hydroxylase (C4H), and 4-coumarate-CoA ligase (4CL). It also increased the accumulation of defense-related substances (total phenol and lignin contents) and up-regulated *AaPAL*, *Aa4CL*, *AaC4H*, and *AaC3′H* gene expression. Furthermore, the combined treatment reduced the disease severity index from 60% to 16% and delayed onset by 2 d. In conclusion, slight water loss combined with MeJA treatment effectively suppressed gray mold. This effect may be attributed to activation of ROS metabolism, induction of phenylpropanoid metabolism, and up-regulation of related genes, which enhanced the resistance of the fruit to gray mold.

## 1. Introduction

*Actinidia arguta* (Siebold & Zucc.) Planch. ex Miq., also known as kiwiberry, baby kiwi, or hardy kiwi, is a representative plant of the genus Actinidia. Its fruits are small, green or dark green, mostly round or oval, sweet, and juicy and have thin, glabrous skin [1]. The pulp has immune-boosting, antioxidant, and antiviral properties [2]. Therefore, this fruit is highly favored by consumers [3]. Notably, *Actinidia arguta* (*A. arguta)* occupies the second-largest area under commercial cultivation in China [4].

High temperatures during the harvest season combined with the rapid metabolism of *A. arguta* lead to postharvest softening of the fruit. Its smooth, hairless skin makes it susceptible to mechanical damage and offers virtually no physical barrier against pathogen invasion. When the fruit peel is damaged during harvesting, transportation, or storage, gray mold fungus (*Botrytis cinerea*), a typical necrotrophic pathogenic fungus, may adhere to the exocarp and invade the flesh tissue, accelerating fruit decay [5,6]. Wounds and senescent tissues serve as the most critical entry points for invasion. As cultivation volumes have increased, the problem of gray mold disease in *A. arguta* during storage has become increasingly severe. The gray mold-infected fruit shows considerably deteriorated sensory quality and storage performance, reducing its commercial value and economic benefits. While chemical control remains the primary method for managing gray mold disease in fruit, it readily induces resistance. Therefore, it is critical to identify methods for preventing and controlling gray mold infection in *A. arguta*.

In recent years, induction of disease resistance in stored fruits using exogenous substances has attracted increasing attention [7,8]. As a natural plant growth regulator, methyl jasmonate (MeJA) is commonly used to enhance the postharvest disease resistance of fruits and vegetables. It effectively induces secondary metabolite production and pathogen resistance by regulating defense-related enzyme activity [9]. Furthermore, extensive studies indicate that MeJA possesses disease-resistance properties. Pre- or postharvest treatment of fruits and vegetables with MeJA effectively suppresses gray mold-induced decay [6]. This effect likely stems from the MeJA-induced accumulation of secondary metabolites, which enhances the synthesis of defense-related substances in the plant, thereby improving postharvest disease resistance [10]. For example, MeJA-treated lychees demonstrate a high sucrose content, improved storage quality, and extended shelf life [9], confirming the role of exogenous MeJA in enhancing disease resistance and preserving fruit quality. Through similar mechanisms, postharvest application of MeJA alleviates cold damage symptoms in papayas [11]. Thus, the effect of MeJA on *A. arguta* disease resistance should be investigated.

In previous studies, we investigated the efficacy of varying degrees of slight water loss against *A. arguta* gray mold infection and its impact on fruit quality. Our preliminary research indicates that a treatment involving 4% water loss can mitigate the occurrence of gray mold disease and reduce postharvest rot in *A. arguta* [12]. This treatment also induces fruit disease resistance by modulating reactive oxygen species (ROS) and phenylpropanoid metabolic pathways. A water loss of 4% yielded the most favorable fruit effects [13]. However, to date, no systematic studies have investigated the relationship between slight water loss, MeJA treatment, and the improvement of disease resistance in A. arguta. Therefore, in this study, we aimed to: (1) compare the inhibitory effects of slight water loss and MeJA treatment on gray mold disease in postharvest *A. arguta*, (2) reval the preliminary disease-resistance induction mechanism of slight water loss combined with MeJA, and (3) investigate the effects of combined treatment on *A. arguta* postharvest disease resistance. To this end, we investigated the combined effects of slight water loss and MeJA treatment on gray mold incidence, antioxidant enzyme (SOD, PPO, and POD) activities, phenylpropanoid metabolism-related enzyme (PAL, C4H, and 4CL) activities, and the expression of related genes (*AaPAL*, *Aa4CL*, *AaC4H*, and *AaC3′H*) in *A. arguta*. Our results offer a conceptual framework and actionable insights for further investigations of the pathogen-defense pathways of *A. arguta*.

## 2. Materials and Methods

### 2.1. Experimental Materials

The *A. arguta* variety “Longcheng II” was selected, with fruits harvested from Kuandian, Dandong, China, in September 2023. At harvest, the fruits showed a uniform shape and size, with a total soluble solids content of 7 ± 0.2% and no visible damage from pests or diseases on their surface. The harvested *A. arguta* fruits were immediately transported to the laboratory within 3 h.

The Botrutis cinerea (*B. cinerea)* strain was sourced from Shenyang Agricultural College, China. The MeJA was purchased from Shanghai McLean Biochemical Technology Co., Ltd. (Shanghai, China).

### 2.2. Preparation of B. cinerea and Spore Suspensions

The *B. cinerea* strain was incubated on potato dextrose agar plates under controlled conditions (26 ± 0.5 °C) for 1 week. The activated pathogenic microorganisms were rinsed with 20 mL of sterile aqueous solution supplemented with 0.05% (*v*/*v*) Tween-20. The concentration was further adjusted to ultimately obtain a standardized concentration of 1 × 10^6^ spores mL^−1^.

### 2.3. Experimental Design and Sample Collection

We applied three different treatments to the *A. arguta* fruits: 4% slight water loss (Treatment 1), MeJA (Treatment 2), and 4% slight water loss + MeJA (Treatment 3) (Table 1). For Treatments 1 and 3, the fruits were subjected to cold air-drying and slight water loss using a laboratory blower. The storage conditions were controlled to keep the temperature at 20 ± 0.5 °C, while the relative humidity (RH) was maintained within the range of 72–78%. Periodic measurements of temperature, humidity, and fruit weight were taken throughout the water-loss process. The treatment was considered complete when the water loss reached 4%. For Treatments 2 and 3, a 10 μmol L^−1^ MeJA solution was evenly sprayed onto the surface of *A. arguta* fruit for 5 min, followed by natural drying. The application of Treatments 1, 2, and 3 was followed by pathogen inoculation; the control group was only subjected to pathogen inoculation. The surface of *A. arguta* fruit was treated with 75% (*v*/*v*) ethanol for disinfection. The fruit was then wounded around the equator with a sterilized needle (5 mm deep), and 10 µL of gray mold spore suspension was introduced into the wound. Each experiment was repeated three times, with three biological replicates per treatment. (Based on preliminary experiments, *Actinidia arguta* exhibits virtually no spontaneous disease development under storage conditions; therefore, this control experiment was omitted.) The experimental design is summarized in Table 1.

After treatment, the *A. arguta* fruits were transferred to storage at ambient temperature (20 ± 0.5 °C, 80–85% RH) for 5 d based on their shelf life, and samples were collected for analysis. Fifteen fruits were randomly selected from each group for testing. Sequential sampling was conducted on Days 0, 1, 2, 3, 4, and 5 of storage to measure the indicators (disease severity index and lesion diameter) of the fresh fruits. Each replicate fruit pulp segment was homogenized separately, then pooled and homogenized (all replicates were processed individually). Samples were immediately frozen using liquid nitrogen and stored at −80 °C for subsequent analysis of total phenolics, lignin content, enzyme activity, and gene expression.

### 2.4. Determination of Disease Severity Index

The diameter of gray mold spots was determined using vernier calipers, and the mean value of the lateral area of the fruit occupied by the surface area of fruit rot was determined over five consecutive days to classify the disease severity index as follows: Grade 0, no decay observed; Grade 1, decay area 0–25%; Grade 2, decay area 26–50%; and Grade 3, decay area > 50%. The disease severity index was calculated as follows:



$$\mathrm{Disease}\;\mathrm{severity}\;\mathrm{index}\;(\%)=\frac{\Sigma \mathrm{decay}\mathrm{level}\times \mathrm{Number}\mathrm{of}\mathrm{fruits}\mathrm{at}\mathrm{this}\mathrm{level}}{\mathrm{Highest}\mathrm{level}\times \mathrm{Total}\mathrm{number}\mathrm{of}\mathrm{fruits}}\times 100\%$$



### 2.5. Assay of POD, PPO, SOD Activities

The POD and PPO activities were determined according to previously reported protocols [14]. Absorbance was measured at wavelengths of 470 nm and 420 nm, respectively, with results expressed as U mg^−1^ protein.

The SOD activity was assessed using an SOD assay kit (ADS-F-KY011; Quanzhou, China), with results expressed as U mg^−1^ protein.

### 2.6. Assay of Lignin and Total Phenol Contents

The lignin content was analyzed according to a previously published method [15], with some modifications. We homogenized 1.0 g of *A. arguta* tissue sample. The pellet was first washed three times with 2 mL of 95% ethanol, followed by three washes with 2 mL of ethanol–hexane solution (1:2 volume ratio). We added 2 mL ice-cold acetic acid, followed sequentially by 100 μL ammonium hydroxide solution (7.5 mol L^−1^), and 1 mL sodium hydroxide solution. After being thoroughly mixed, the solution was centrifuged at 8000× *g* for 25 min at 4 °C. Subsequently, 400 μL of the supernatant was mixed with 4.6 mL of acetic acid, and the absorbance was measured at a wavelength of 280 nm. The results were expressed as OD_280_ g^−1^ kg^−1^.

The total phenol content was assayed using a modified version of a previously published method [16]. We homogenized 1.0 g of *A. arguta* tissue sample in 80% ethanol, added distilled water, NaCO_3_, and Folin reagent; and mixed them thoroughly. The supernatant was collected for analysis, and the absorbance was measured at a wavelength of 760 nm.

### 2.7. Assay of PAL, 4CL, and C4H Activities

The PAL, 4CL, and C4H activities were determined using previously established methods [17,18]. Absorbance was measured at wavelengths of 290, 330, and 340 nm, respectively, and remeasured after incubation in a water bath at 40 °C for 1 h. Results were expressed as U mg^−1^ protein.

### 2.8. Assay of the Expression of Genes Corresponding to Key Enzymes in Phenylalanoid Metabolism

We extracted the total RNA from *A. arguta* tissues using a cDNA synthesis kit (BioWorks, Victor, NY, USA). Analysis using geNorm and NormFinder (Microsoft Excel 365) software ultimately identified Actin as the optimal housekeeping gene. Gene sequences encoding key phenylpropanoid metabolic enzymes in *Actinidia arguta*—*AaPAL*, *Aa4CL*, *AaC4H*, and *AaC3′H*—were obtained. Specific primers were selected using the National Center for Biotechnology Information database and synthesized through bioengineering (Table 2). We performed real-time qPCR amplification using the UltraSYBR Mixture (Jiangsu Kangwei Century Biotechnology Co., Ltd., Taizhou, China). Each experiment used three cDNA samples, each of which was prepared from three biological replicates. The comparative 2^−ΔΔCT^ method was employed to compute relative expression levels of target genes [19].

### 2.9. Statistical Analyses

All experimental results are based on three independent biological and technical replicates. Each sample underwent at least three parallel analytical determinations. All data are presented as the mean and its corresponding standard deviation. Univariate analysis of variance was performed using SPSS v23 software. Intergroup comparisons were conducted using Duncan’s multiple range test, with *p* < 0.05 as the statistical significance threshold. All graphs were plotted using Origin 2018.

## 3. Results

### 3.1. Effects of Different Treatments on the Disease Severity Index

Lesion diameter and the disease severity index are two key indicators employed to assess the severity of fruit diseases. As shown in Figure 1, the disease severity index and lesion diameters of *A. arguta* fruit exhibited gradual increases in both the control and treatment groups during storage. Treatment 3 significantly reduced the disease severity index of inoculated fruit (*p* < 0.05). On Day 5 after inoculation, the disease severity index of *A. arguta* fruit was 40%, 43%, and 16% under Treatments 1, 2, and 3, respectively, representing decreases of 20%, 17%, and 44% compared with that under the control (60%), respectively. The diameters of the lesions were 19 mm, 20 mm, and 11 mm under Treatments 1, 2, and 3, respectively, which were 24%, 20%, and 56% smaller than that under the control, respectively. Overall, Treatment 3 exhibited the strongest inhibitory effect against gray mold infection in *A. arguta* fruit that had been inoculated with the pathogen.

### 3.2. Effects of Different Treatments on the POD, PPO, and SOD Activities

In both the control and treatment groups, the activities of ROS-metabolizing enzymes (including POD, PPO, and SOD) in *B. cinerea*-inoculated *A. arguta* showed a trend of first increasing and then decreasing as the storage period progressed.

The POD and PPO activities in pathogen-inoculated *A. arguta* showed similar trends. Treatments 1 and 3 showed essentially the same trend as the control, after peaking on Day 4, the enzyme activity began to decline. (Figure 2A,B). The enzyme activity of *A. arguta* under all three treatments showed a similar trend to that under the control. However, the POD and PPO activities under the combined treatment were significantly higher than those under the other treatments (*p* < 0.05). The POD activity was 68.78 U mg^−1^ protein higher under Treatment 3 than under the control (93.4 U mg^−1^ protein). The PPO activity was 68.74 U mg^−1^ protein higher under Treatment 3 than under the control (91.4 U mg^−1^ protein).

The SOD activity exhibited the same trend across all treatment groups as in the control group. It increased from Day 0 to Day 1, peaked on Day 3, and then began to decline (Figure 2C). The combined treatment group showed significantly higher SOD activity (1.67 U mg^−1^ protein) than the control group (2.61 U mg^−1^ protein) (*p* < 0.05) on Day 5 after inoculation.

Overall, the combined treatment resulted in a significant increase in POD, PPO, and SOD activities. These three enzymes may decompose hydrogen peroxide and participate in lignin synthesis, thereby reducing damage to fruit cells while strengthening the physical barrier of cell walls, thus enhancing disease resistance.

### 3.3. Effects of Different Treatments on the Total Phenol and Lignin Contents

In both the control and treatment groups, the total phenol content in inoculated *A. arguta* exhibited a trend of an initial increase followed by a subsequent decrease (Figure 3A). On Day 3 after inoculation, the total phenolic contents under Treatments 1, 2, and 3 were 9.98%, 23.03%, and 25.46% higher than that under the control, respectively, indicating that the combination of 4% slight water loss and MeJA treatment led to the greatest increase in the total phenolic content of *A. arguta* during storage.

Lignin accumulation in the pathogen-inoculated *A. arguta* also increased with storage time (Figure 3B). One day after inoculation, notable differences in lignin content were detected between the three treatment groups and the control group (*p* < 0.05), with the values under Treatments 1, 2, and 3 being 0.278, 0.282, and 0.288 OD_280_ g^−1^ kg^−1^, respectively. Throughout the entire experimental period, Treatment 3 consistently had the highest lignin content among all groups. This indicates that the combined treatment promoted the synthesis of lignin, which was effective against infection by gray mold fungus.

### 3.4. Effects of Different Treatments on PAL, 4CL, and C4H Activities

The enzymes PAL, 4CL, and C4H are closely associated with phenylpropanoid metabolism in plant secondary metabolic pathways. The PAL activity in the pathogen-inoculated *A. arguta* showed a gradually increasing trend during storage (Figure 4A). However, the PAL activity was significantly higher in the three treatment groups than in the control group (*p* < 0.05). On Day 5 after inoculation, it increased by 38.37%, 32.74%, and 40.97% under Treatments 1, 2, and 3, respectively, compared with that under the control.

The trends in the C4H and PAL activities were similar in the control and treatment groups (Figure 4C). On Day 5 after inoculation, the C4H activity significantly increased by 38.37%, 32.74%, and 40.98% under Treatments 1, 2, and 3 (*p* < 0.05), respectively, compared with that under the control.

The 4CL activity showed an overall trend of an initial increase followed by a decline (Figure 4B). On Day 5 after inoculation, the 4CL activity significantly increased by 35.71%, 27.14%, and 37.23% under Treatments 1, 2, and 3 (*p* < 0.05), respectively, compared with that under the control. The activities of these enzymes (PAL, 4CL, and C4H) were consistently the highest under the combined treatment.

### 3.5. Effects of Different Treatments on the Expression of Genes Related to Key Enzymes of Phenylpropanoid Metabolism

The relative expressions of *AaPAL*, *AaC4H*, *AaC3′H* and *Aa4CL* in the inoculated fruits showed trends of first increasing and then decreasing (Figure 5). The relative expression of *AaPAL* peaked on Day 3 after inoculation and then decreased. On Days 3–4 after inoculation, the level of relative expression was 18.6% higher under Treatment 3 than under the control (*p* < 0.05), and it remained at the highest level during storage.

The relative expression of *AaC4H* showed a similar trend across the control group and all treatment groups. On Days 2–5 after inoculation, the relative expression level of AaC4H in the combined treatment group was 74.5% higher than that in the control group (*p* < 0.05).

There was a rapid increase in the relative expression of *AaC3′H* on Days 1–4. The relative expression level of *AaC3′H* was significantly higher under Treatment 3 than under the control on Day 4 after inoculation (*p* < 0.05).

The expression of *Aa4CL* showed similar trends in the control and treatment groups. On Days 1–5 after inoculation, the relative expression levels of *Aa4CL* in all three treatment groups exceeded that in the control group, with the combined treatment group showing a 40% higher expression level compared with that in the control group (*p* < 0.05). The combined treatment significantly increased the relative expression levels of *AaPAL*, *AaC4H*, *AaC3′H* and *Aa4CL* in the fruit (*p* < 0.05). The increase in the relative expression levels of *AaPAL*, *AaC4H*, and *Aa4CL* in the combined treatment group corresponded to the synchronous increase in the PAL, C4H, and C3′H enzyme activities, as well as to the increase in total phenolic content.

## 4. Discussion

Currently, chemical application is the main method for gray mold control owing to its low cost and rapid effects [20,21]. However, the long-term use of chemicals may induce resistance in pathogenic bacteria. Furthermore, pesticide residues may remain on the surface of fruits, affecting the nutrient composition and sensory quality as well as posing potential food safety risks. Therefore, the development of safe, natural control technologies has gained increasing attention in the prevention and control of gray mold.

As a naturally occurring plant signaling compound, MeJA plays a primary role in activating innate defense mechanisms against both biotic and abiotic stressors. Numerous studies have demonstrated that jasmonic acid treatment effectively reduces disease incidence and improves fruit and vegetable quality. For example, this treatment has been effective in kiwifruit [21], grapes [22], bayberries [23], and cherry tomatoes [24]. Additionally, abiotic stressors (e.g., water stress) can stimulate fruits to produce defense-related substances and activate defense enzymes, thereby enhancing pathogen resistance. Such induced resistance exhibits broad-spectrum and persistent characteristics [25].

Preliminary laboratory studies indicated that slight water loss enhances fruit disease resistance [12]. In the current study, we found that both slight water loss and MeJA treatment individually demonstrated comparable disease-suppression effects. Consistent with previous research findings [13], MeJA treatment reduced fruit disease incidence, with no significant difference between slight water loss and MeJA treatment. Furthermore, the disease-resistance mechanism induced by slight water loss was similar to that induced by MeJA treatment, and both methods reduced gray mold incidence by inducing disease resistance. Notably, the combined treatment (4% slight water loss + 10 μmol L^−1^ MeJA) produced a synergistic effect on induced resistance. It significantly reduced the incidence of gray mold, potentially by inducing defense-related enzyme activity to enhance kiwifruit disease resistance.

Similar results have been found in other fruits. Postharvest loquats treated with MeJA showed a significant (*p* < 0.05) reduction in anthracnose incidence [26]. Increased lignin content following MeJA treatment in longans significantly improved their storage quality and disease resistance [27]. Notably, phenylpropanoid metabolism provides substrates for lignin formation and produces phenolic compounds with antifungal and antioxidant activities. Total plant phenols can also be converted into lignin. Treatment with 200 μmol L^−1^ MeJA was shown to increase the flavonoid and total phenolic contents in apples, resulting in the control of postharvest penicillium mold [28]. Our experiment demonstrated that *A. arguta* treated with MeJA showed significantly reduced disease incidence and lesion diameter, with increased fruit lignin and total phenolic contents (*p* < 0.05). Slight water loss produced effects comparable to those of MeJA treatment, with both treatments showing markedly superior disease-resistance induction effects compared with those of the control.

Highly active oxygen-containing molecules (i.e., ROS) are produced during plant cell metabolism. When plants encounter biotic stressors such as pathogen infection, the antioxidant system composed of enzymes, including SOD, POD, and PPO, forms the core mechanism for regulating ROS metabolic balance and mitigating oxidative damage [29]. Thus, SOD, POD, and PPO are crucial antioxidant enzymes in plant disease resistance, participating in the response to abiotic stress and protecting fruits and vegetables from ROS damage [30,31]. Illustrative examples include saints’ fruit [32], pear [33], and kiwifruit [34]. Our results indicate that MeJA treatment increased the activities of POD, SOD, and PPO compared with those under the control. Consistent findings have been reported in previous studies [2]. These results further confirm that the disease resistance induced by slight water loss is associated with enhanced antioxidant enzyme activity in *A. arguta*.

In the phenylpropanoid pathway, PAL is a crucial enzyme that transforms phenylalanine into trans-cinnamic acid. The production of lignin and phenolics, which are directly linked to fruit disease resistance, is facilitated by three essential enzymes (PAL, C4H, 4CL) in the phenylpropanoid pathway [35,36]. The synergistic interaction of these three components forms the core metabolic pathway: PAL → C4H → 4CL. The intermediate product catalyzed by 4CL polymerizes under the action of downstream enzymes to form lignin, enhancing the mechanical strength and degradation resistance of cell walls to form a physical barrier to infection. It also synthesizes secondary metabolites such as total phenolics and flavonoids, which exhibit potent antibacterial activity [36,37]. In the present study, the 4% water loss and MeJA treatment resulted in fruit disease resistance and rapid enzyme response (*p* < 0.05). The peaks of PAL, C4H, and 4CL enzyme activities occurred earlier under slight water loss alone than under MeJA treatment alone. These results are consistent with those of other preservation treatments [13,37,38,39].

We also found that the combinations of slight water loss and MeJA treatment significantly outperformed both the control and individual treatments (*p* < 0.05). This combination reduced fruit rot and enhanced the accumulation of defense-related enzymes associated with ROS and phenylpropanoid metabolism. In this study, we observed synchronized changes in enzyme activity and phenolic compounds, but the causal relationship between the two and the specific regulatory mechanisms require further validation through experiments such as gene silencing and in vitro enzymatic reactions.

To further verify whether the resistance to *B. cinerea* induced by the combined treatment was linked to the activation of the defense response in *A. arguta*, real-time qPCR analysis was conducted. Analysis of gene expression levels revealed that slight water loss combined with MeJA treatment elevated the relative expression of the *AaPAL*, *Aa4CL*, *AaC4H*, and *AaC3′H* genes. These findings are aligned with the results of other preservation treatments [21,40]. Notably, these genes were expressed in the control group, and both slight water loss and MeJA treatment alone increased the relative expression levels of these genes, but their expression levels were considerably higher in the combined treatment group. Therefore, slight water loss exhibited disease-resistance effects similar to those of MeJA treatment, whereas the combined treatment demonstrated a more pronounced synergistic effect, outperforming either treatment alone. Overall these results indicate that activation of ROS metabolism and phenylpropanoid pathway-associated enzyme activity during storage effectively controls gray mold disease in *A. arguta* while preserving high fruit quality. This study demonstrates that postharvest water loss and MeJA application can be used to prevent postharvest diseases in fruits and vegetables. The specific mechanisms underlying the synergistic effects of water loss and MeJA remain largely unexplored, warranting future research into the detailed mechanisms of this treatment combination.

## 5. Conclusions

We confirmed that slight water loss and MeJA treatment share a similar mechanism of action. Furthermore, we found that slight water loss combined with MeJA treatment increased the resistance of *A. arguta* fruits to gray mold, which may be closely related to the activation of ROS and phenylpropanoid metabolism. This combined treatment increased the activity of enzymes related to ROS and phenylpropanoid metabolism, increased the content of secondary metabolites, and up-regulated the key enzyme genes for phenylpropanoid metabolism. Combined water loss and MeJA treatment demonstrated significant efficacy against gray mold disease in *Actinidia arguta* by inducing fruit disease resistance and enhancing the synthesis of defense-related substances. However, its effectiveness in field applications effectiveness and adaptability across different cultivars and storage conditions require further validation.

## Figures and Tables

**Figure 1 foods-14-04311-f001:**
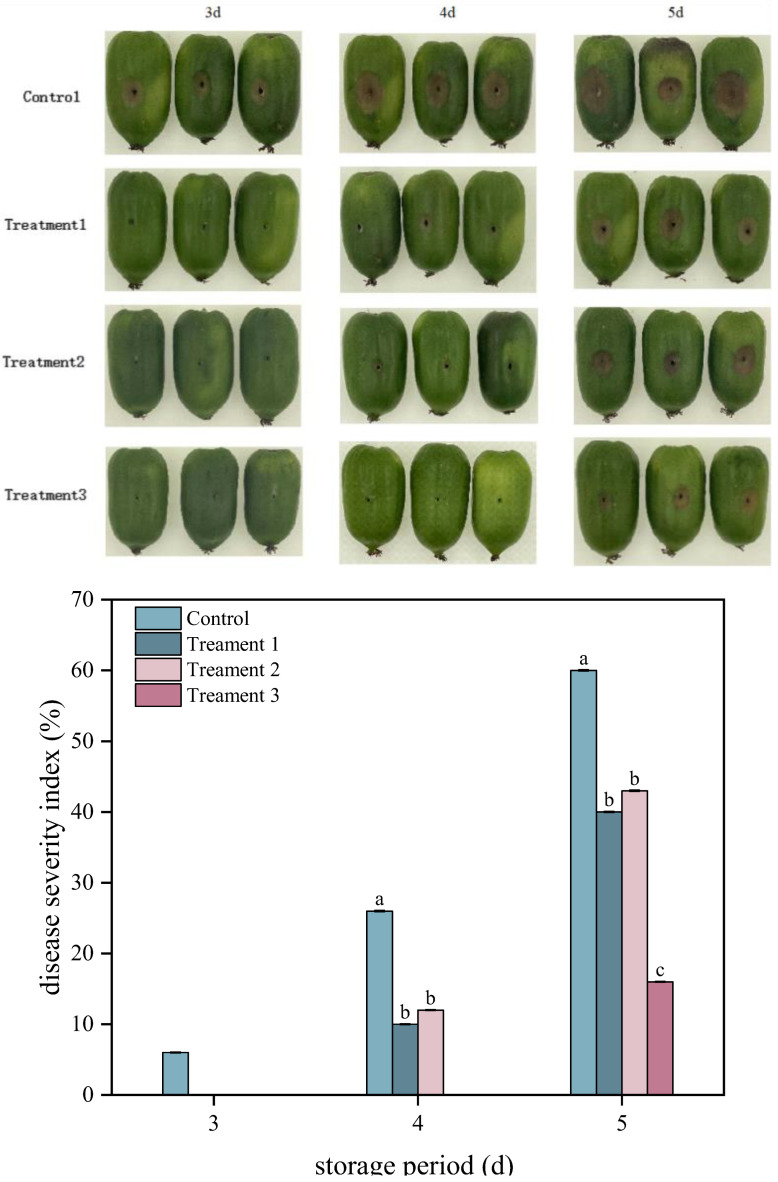
Effects of different treatments disease severity index and symptoms of *Actinidia arguta*. Statistical significance was designated as *p* < 0.05. Different lowercase letters (a–c) indicate significant differences between treatments at *p* < 0.05.

**Figure 2 foods-14-04311-f002:**
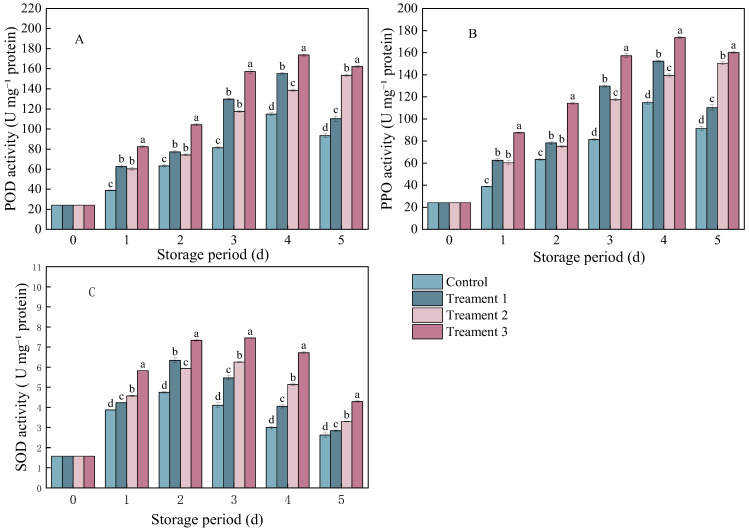
The activities of (**A**) peroxidase (POD), (**B**) polyphenol oxidase (PPO), and (**C**) superoxide dismutase (SOD) in *A. arguta* infected with gray mold and subjected to different treatments. Statistical significance was designated as *p* < 0.05. Different lowercase letters (a–d) indicate significant differences between treatments at *p* < 0.05.

**Figure 3 foods-14-04311-f003:**
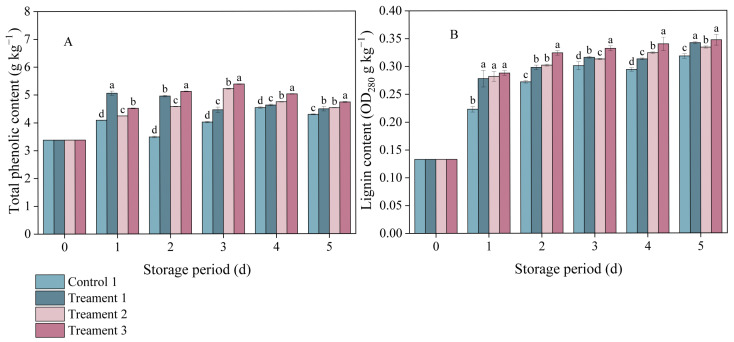
The effects of different treatments on (**A**) total phenolic and (**B**) lignin contents. Statistical significance was designated as *p* < 0.05. Different lowercase letters (a–d) indicate significant differences between treatments at *p* < 0.05.

**Figure 4 foods-14-04311-f004:**
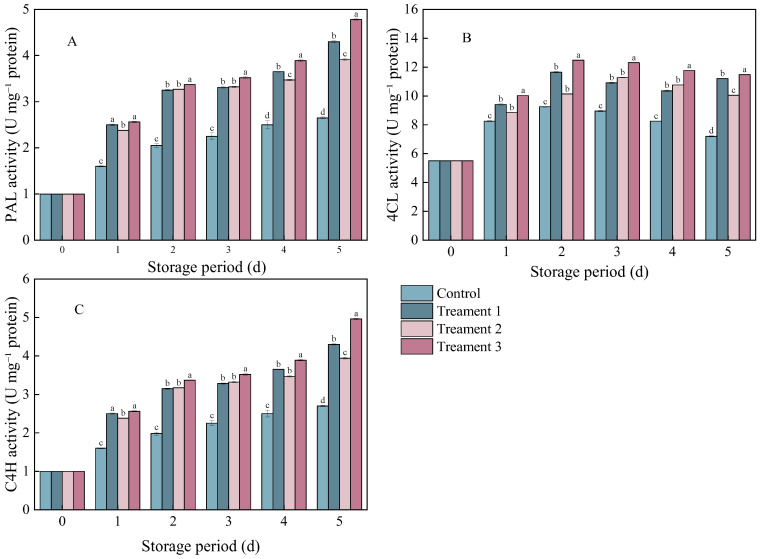
The effects of different treatments on (**A**) phenylalanine ammonia-lyase (PAL), (**B**) 4-coumarate-CoA ligase (4CL), and (**C**) cinnamate 4-hydroxylase (C4H) activities. Statistical significance was designated as *p* < 0.05. Different lowercase letters (a–d) indicate significant differences between treatments at *p* < 0.05.

**Figure 5 foods-14-04311-f005:**
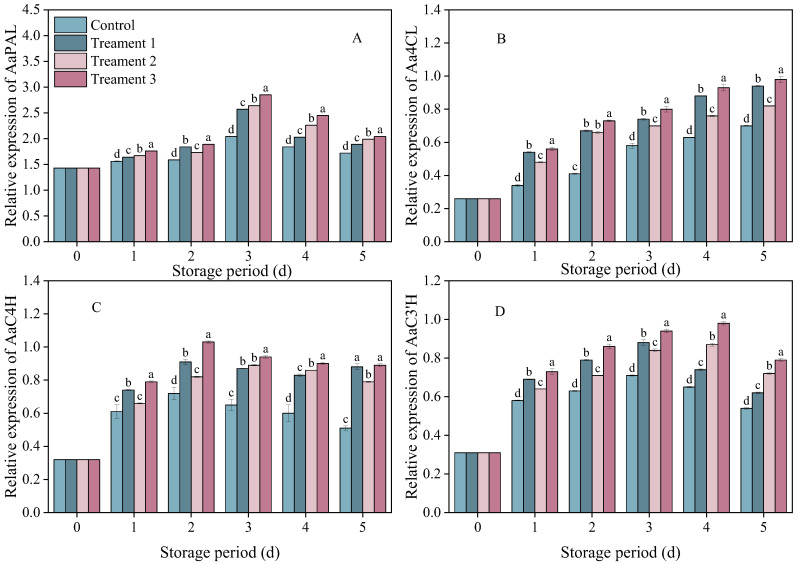
The effects of the different treatments on the expression of (**A**) *AaPAL*, (**B**) *Aa4CL*, (**C**) *AaC4H*, and (**D**) *AaC3′H*. Statistical significance was designated as *p* < 0.05. Different lowercase letters (a–d) indicate significant differences between treatments at *p* < 0.05.

**Table 1 foods-14-04311-t001:** The specific processing methods employed in this study.

Treatment	Control	Treatment 1	Treatment 2	Treatment 3
4% water loss		√		√
MeJA			√	√
pathogen	√	√	√	√

**Table 2 foods-14-04311-t002:** Primers used for real-time q-PCR analysis.

Gene	Primer Sequence 5′ → 3′
*Actin*	F: GTGCTCAGTGGTGGTTCAA
	R: GACGCTGTATTTCCTCTCAG
*AaPAL*	F: AACCGGATTAAGGAGTGCCG
	R: GGTGACTGCACCTTCTCTCC
*Aa4CL*	F: AGTCGAAATCAGCCCAGACG
	R: GTCGCTGTGCATGTAGAGGT
*AaC4H*	F: GACACCCAAAAGCTCCCGTA
	R: CTCTGCGGGGATGTCGTATC
*AaC3′H*	F: ACCGAATGGTCTCAAGCCAG
	R: TCCACAGGCACACGTTTGTA

## Data Availability

The original contributions presented in the study are included in the article, further inquiries can be directed to the corresponding author.

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
