# Peer review of "Slight Water Loss Combined with Methyl Jasmonate Treatment Improves *Actinidia arguta* Resistance to Gray Mold by Modulating Reactive Oxygen Species and Phenylpropanoid Metabolism"

_foods, 2025, doi:10.3390/foods14244311_

Round 1

Reviewer 1 Report

Comments and Suggestions for Authors

Title:

 SUGGESTION: "Water stress combined with methyl jasmonate enhances Actinidia arguta resistance to Botrytis cinerea via ROS and phenylpropanoid metabolism"

ABSTRACT (Lines 5-22)

CRITICAL ERRORS:

 "phenylpropane metabolism"

  • CORRECTION: "phenylpropanoid metabolism"

"incidence rate"

  • PROBLEM: Manuscript formula (line 138) calculates severity index, NOT incidence
  • CORRECTION: Either:
    • Use true incidence formula AND report actual values
    • OR rename to "disease severity index" and specify scale
  • CURRENT CLAIM: "73.33% lower" - meaningless without baseline values
  • REQUIRED: Report absolute values (e.g., "decreased from 60% to 16%")

"The results showed that the three treatments"

  • UNCLEAR: Which three treatments? Reader doesn't know yet.
  • SUGGESTION: "Water loss alone, MeJA alone, and their combination all reduced disease, with combined treatment showing greatest efficacy"

Line 22: "A. arguta gray mold infection"

  • IMPRECISE: "Botrytis cinerea infection" is more specific and scientifically accurate

Missing from abstract:

  • Sample size (n=?)
  • Storage duration (how many days evaluated?)
  • Statistical significance indicators
  • Key numerical findings (specific % reductions, fold-changes)
  • Reduce methods to 2-3 lines, expand results with quantitative data

SUGGESTED KEYWORDS:

  • Actinidia arguta
  • Botrytis cinerea
  • Postharvest water stress
  • Methyl jasmonate
  • Phenylpropanoid metabolism
  • Reactive oxygen species
  • Induced resistance

INTRODUCTION (Lines 25-95)

Line 33: "Actinidia arguta, also known as miniature kiwifruit"

  • PROBLEM: Missing author citation for species name
  • CORRECTION: "Actinidia arguta (Siebold & Zucc.) Planch. ex Miq., also known as kiwiberry, baby kiwi, or hardy kiwi"

Vague disease description

: Gray mold infection mechanism description

  • PROBLEM: Oversimplified. No mention that B. cinerea is necrotrophic, requires wounds/senescence
  • MISSING: "Botrytis cinerea is a necrotrophic fungal pathogen..."
  • REQUIRED: Specify that damage during harvest/handling creates infection courts

Weak justification for study

Lines 89-95: Objective statement

  • PROBLEM: Does not explicitly state hypothesis about synergism mechanism
  • MISSING: "We hypothesized that slight water stress would prime ABA-mediated defenses, which MeJA treatment would subsequently amplify through JA pathway activation, resulting in synergistic protection"

NEEDED: Expand to 2-3 sentences covering:

  • B. cinerea host range (polyphagous, >200 hosts)
  • Infection requirements (wounds, senescent tissue)
  • Environmental factors (humidity, temperature)
  • Current control limitations (fungicide resistance)

Lines 53-58 - Water loss mechanism unclear

Current: States water loss enhances resistance but mechanism vague

NEEDED: Specify that water stress triggers ABA accumulation and primes defense responses

Lines 64-73 - MeJA mechanism superficial

Current: Lists several defense responses activated by MeJA

NEEDED:

  • Specify COI1-JAZ-MYC2 signaling cascade
  • Mention JA-responsive genes (PDF1.2, VSP, LOX)
  • Connect to phenylpropanoid pathway specifically

Lines 74-88 - Phenylpropanoid pathway description inadequate

Current: General overview of pathway

 PROBLEMS:

  • No mention of entry point (PAL converts phenylalanine → cinnamic acid)
  • Doesn't specify that C4H and 4CL are core enzymes
  • Missing link between lignin and physical barrier formation
  • No connection to antimicrobial properties

Objective lacks specificity

Current: General statement about investigating relationships

 PROBLEMS:

  • Doesn't specify which defense components will be measured
  • No hypothesis about synergism mechanism
  • Missing specific aims

RESULTS SECTION (Lines 196-326)

Disease Data Reporting - Lines 196-210

FUNDAMENTAL PROBLEM: The "incidence" data reported are actually severity indices, as identified in the incorrect methodology (line 138).

CRITICAL ABSENCES:

  • Initial and final absolute values not clearly provided
  • Magnitude of differences between treatments (fold-change, %) inconsistent
  • Lacks day zero baseline (post-inoculation)

SOD Activity Data - Lines 211-223

PROBLEM: Confusing presentation, relative values without absolute baseline, superficial interpretation.

Line 214-216: "a rapid decline under Treatment 1, and a consistent increase under Treatment 2"

INADEQUACIES:

  • Does not provide absolute activity values (U mg⁻¹ protein)
  • Only percentages relative to control
  • Lacks magnitude of changes (fold-change)
  • Does not explain biological significance of variations

POD and PPO Activities - Lines 224-245

SIMILAR PROBLEMS: Lack of absolute values, mechanical interpretation, without connection to biological meaning.

Phenolic Metabolites - Lines 246-274

Total phenols data based on incorrect methodology (325 nm instead of 760 nm - see Methods, line 162).

Enzyme Gene Expression - Lines 275-306

MULTIPLE PROBLEMS:

  • Lacks RT-qPCR validation (efficiency, reference stability)
  • Fold-change values not numerically provided
  • Only visual description of figures
  • Without gene-enzyme-metabolite correlation analysis

PAL, C4H, 4CL Enzyme Activities - Lines 307-326

PROBLEMS:

  • Fragmented and disorganized presentation
  • Lacks integration among the three pathway enzymes
  • Does not discuss biological significance of enzymatic sequence
  • Without correlation with final products (lignin)

DISCUSSION SECTION (Lines 327-407)

It was verified that the Discussion section attempts to contextualize experimental findings within existing literature on induced resistance mechanisms in fruit postharvest systems. However, critical deficiencies were identified in mechanistic interpretation, particularly regarding the synergistic interaction between water stress and methyl jasmonate, which constitutes the central novelty claim of this manuscript. The discussion presents superficial, repetitive content lacking the depth of analysis

DEFICIENCY 1: Synergism Mechanism NOT Explained (Lines 327-407)

FUNDAMENTAL PROBLEM: The manuscript's central claim - that water loss and MeJA produce synergistic (not merely additive) effects - is never mechanistically explained.

WHAT IS MISSING: The discussion completely fails to address the molecular mechanism underlying synergistic interaction. No mention is made of:

  • ABA-JA hormonal crosstalk pathways
  • Shared or convergent transcription factors (MYC2, ABF, AREB)
  • Priming concept (water stress sensitizes defense responses)
  • MAPK signaling cascade convergence
  • Chromatin remodeling or epigenetic mechanisms

DEFICIENCY 2: Superficial Literature Integration (Throughout)

PROBLEM: Citations are used mechanically to state "others found similar results" without critical analysis, comparison of magnitudes, or discussion of discrepancies.

DEFICIENCY 3: Lack of Critical Evaluation of Own Results

PROBLEM: The discussion accepts all results uncritically without acknowledging limitations, unusual patterns, or unexpected findings.

DEFICIENCY 5: Inadequate Discussion of Practical Implications

PROBLEM: The discussion fails to address how findings could be translated to commercial practice, economic feasibility, or scalability.

DEFICIENCY 7: Incorrect/Repeated Nomenclature

CRITICAL ERROR: "phenylpropane pathway" instead of "phenylpropanoid pathway"

CORRECTION REQUIRED: Replace ALL instances of "phenylpropane" with "phenylpropanoid"

MISSING 2: Species-Specific Considerations

NOT DISCUSSED: Why A. arguta responses might differ from other fruits.

fruit species.

MISSING 3: Future Research Directions

 The discussion fails to identify specific hypotheses or experiments needed to address knowledge gaps.

CONCLUSIONS SECTION (Lines 408-417)

It was verified that the Conclusions section presents a brief synthesis of main experimental findings. However, critical deficiencies were identified in scope, depth, and scientific rigor. The conclusions are excessively simplistic, fail to articulate the study's novel contribution, lack appropriate cautious language regarding limitations, and do not provide actionable recommendations for future research or practical application.

Conclusions provide no guidance on how findings could be applied commercially or what conditions are required for practical implementation.

Conclusions fail to identify key knowledge gaps or propose specific next steps for advancing the field.

DEFICIENCY 5: Weak Final Statement (Line 417)

Line 417: " The results of this  study provide a theoretical foundation for improved postharvest storage of A. arguta."

PROBLEMS:

  1. Vague and generic - Could be said about any postharvest disease study
  2. No specific actionable insight - What exactly should be done with these findings?
  3. Missed opportunity to highlight unique contribution of demonstrating synergism
  4. No forward-looking vision for the field

PROBLEM: Conclusions don't clearly state what is novel in this study versus what confirms previous findings.

< !--a=1-->

Author Response

1、CRITICAL ERROR: "phenylpropane pathway" instead of "phenylpropanoid pathway"

The author’s answer:

We sincerely appreciate the valuable comments. We have carefully reviewed the entire document and made revisions in red font in the revised draft.

2、PROBLEM: Manuscript formula (line 138) calculates severity index, NOT incidence

The author’s answer:

We sincerely appreciate the valuable comments. We have made the change to “disease severity index” in the revised draft.

3、CURRENT CLAIM: "73.33% lower" - meaningless without baseline values. Which three treatments? Reader doesn't know yet. Line 22: "A. arguta gray mold infection IMPRECISE.SUGGESTED KEYWORDS.

The author’s answer:

We sincerely appreciate the valuable comments. We have increased Furthermore, the combined treatment reduced the disease severity index from 60% to 16% and delayed onset by 2 d. (Line 24-25). We have detailed the three processing methods as follows:water loss alone, MeJA alone, and their combination each reduced the incidence of disease, with the combined treatment showing the greatest efficacy. We have revised the abstract and keywords in the draft based on your suggestions. We have changed the keywords to Actinidia arguta, Botrytis cinerea, Postharvest water stress, Methyl jasmonate, Phenylpropanoid metabolism, Reactive oxygen species, and Induced resistance (Line 30-31).

4、 Missing author citation for species name. Oversimplified. No mention that B. cinerea is necrotrophic, requires wounds/senescence.

The author’s answer:

We sincerely appreciate the valuable comments. We have changed Actinidia arguta, also known as miniature kiwifruit, to Actinidia arguta (Siebold & Zucc.) Planch. ex Miq., also known as kiwiberry, baby kiwi, or hardy kiwi(Line37-38). Regarding the description of the gray mold infection mechanism, we have added the following in the revised draft:When the fruit peel is damaged during harvesting, transportation, or storage, gray mold fungus (Botrytis cinerea), a typical necrotrophic pathogenic fungus, may adhere to the exocarp and invade the flesh tissue, accelerating fruit decay[5]. Wounds and senescent tissues serve as the most critical entry points for invasion.(Line 46-49).

5、Does not explicitly state hypothesis about synergism mechanism. Lines 53-58 - Water loss mechanism unclear. Lines 64-73 - MeJA mechanism superficial.  

The author’s answer:

Thank you for your valuable suggestions. Indeed, when designing the experiment, we were not yet clear on the mechanisms underlying slight water loss. We focused on the impact of slight water loss on disease susceptibility, as discussed in our published paper “Slight water loss affects the quality of ‘Longcheng 2’ kiwiberry fruit infected with gray mold disease,” and attempted to validate its mechanisms through the regulation pathway of methyl jasmonate (MeJA). Interestingly, we discovered a synergistic effect between the two, prompting us to initiate a series of follow-up studies. Moving forward, we will delve deeper into how the mechanisms of slight water loss and MeJA interact to influence fruit disease resistance, aiming to develop a more comprehensive understanding of this phenomenon.

6、Lines 74-88 - Phenylpropanoid pathway description inadequate

The author’s answer:

Thank you for your valuable suggestions. I have added this section to the discussion, with the following content: Line 377-386, In the phenylpropanoid pathway, PAL is a crucial enzyme that transforms phenylalanine into trans-cinnamic acid. The production of lignin and phenolics, which are directly linked to fruit disease resistance, is facilitated by three essential enzymes (PAL, C4H, 4CL) in the phenylpropanoid pathway[36,37]. The synergistic interaction of these three components forms the core metabolic pathway: PAL→C4H→4CL. The intermediate product catalyzed by 4CL polymerizes under the action of downstream enzymes to form lignin, enhancing the mechanical strength and degradation resistance of cell walls to form a physical barrier to infection. It also synthesizes secondary metabolites such as total phenolics and flavonoids, which exhibit potent antibacterial activity[36,37].

7、Disease Data Reporting - Lines 196-210. Initial and final absolute values not clearly provided. Magnitude of differences between treatments (fold-change, %) inconsistent. Lacks day zero baseline (post-inoculation).

The author’s answer:

Thank you for your valuable suggestions. We have made the following modifications as requested: Line 211-222, Lesion diameter and the disease severity index are two key indicators employed to assess the severity of fruit diseases. As shown in Fig. 1, the disease severity index and lesion diameters of A. arguta fruit exhibited gradual increases in both the control and treatment groups during storage. Treatment 3 significantly reduced the disease severity index of inoculated fruit (P < 0.05). On Day 5 after inoculation, the disease severity index of A. arguta fruit was 40%, 43%, and 16% under Treatments 1, 2, and 3, respectively, representing decreases of 20%, 17%, and 44% compared with that under the control (60%), respectively. The diameters of the lesions were 19 mm, 20 mm, and 11 mm under Treatments 1, 2, and 3, respectively, which were 24%, 20%, and 56% smaller than that under the control, respectively. Overall, Treatment 3 exhibited the strongest inhibitory effect against gray mold infection in A. arguta fruit that had been inoculated with the pathogen. The absence of data for 0 days is because the four treatments were identical at day 0 with no discernible differences, so we did not collect data. During storage from 0 to 2 days, no sensory differences emerged, hence no data collection. Day 5 represents the final experimental value.

8、SOD Activity Data - Lines 211-223.  Confusing presentation, relative values without absolute baseline, superficial interpretation. POD and PPO Activities - Lines 224-245.: Lack of absolute values, mechanical interpretation, without connection to biological meaning.

The author’s answer:

Thank you for your valuable suggestions. We have converted the measurement results to data expressed in U mg-1 protein.

9、Total phenols data based on incorrect methodology (325 nm instead of 760 nm - see Methods, line 162)

The author’s answer:

We appreciate the reviewers' insightful comments! The total phenolic content data has been corrected based on the re-measurement of the method, and the new results have been incorporated into the revised manuscript.

10、Enzyme Gene Expression - Lines 275-306

The author’s answer:

We appreciate the reviewers' insightful comments! We revised this section to read as follows: Line 172-181, We extracted the total RNA from A. arguta tissues using a cDNA synthesis kit (BioWorks). Analysis using geNorm and NormFinder software ultimately identified Actin as the optimal housekeeping gene. Gene sequences encoding key phenylpropanoid metabolic enzymes in Actinidia arguta—AaPAL, Aa4CL, AaC4H, and AaC3'H—were obtained. Specific primers were selected using the National Center for Biotechnology Information database and synthesized through bioengineering (Table 2). We performed real-time qPCR amplification using the UltraSYBR Mixture (Jiangsu Kangwei Century Biotechnology Co., Ltd., China). Each experiment used three cDNA samples, each of which was prepared from three biological replicates. The comparative 2-ΔΔCT method was employed to compute relative expression levels of target genes[19]. The enzyme gene section includes data descriptions and adds descriptions of the correlations between enzymes, genes, and phenolic compounds. We revised this section to read as follows: Line 300-320, The relative expressions of AaPAL, AaC4H, AaC3'H and Aa4CL in the inoculated fruits showed trends of first increasing and then decreasing (Fig. 5). The relative expression of AaPAL peaked on Day 3 after inoculation and then decreased. On Days 3–4 after inoculation, the level of relative expression was 18.6% higher under Treatment 3 than under the control (P < 0.05), and it remained at the highest level during storage.

 On Days 2–5 after inoculation, the relative expression level of AaC4H in the combined treatment group was 74.5% higher than that in the control group (P < 0.05).

On Days 1–5 after inoculation, the relative expression levels of Aa4CL in all three treatment groups exceeded that in the control group, with the combined treatment  group showing a 40% higher expression level compared with that in the control group (P < 0.05). The combined treatment significantly increased the relative expression levels of AaPAL, AaC4H, AaC3'H and Aa4CL in the fruit (P < 0.05). The increase in the relative expression levels of AaPAL, AaC4H, and Aa4CL in the combined treament group corresponded to the synchronous increase in the PAL, C4H, and C3'H enzyme activities, as well as to the increase in total phenolic content.

11、PAL, C4H, 4CL Enzyme Activities - Lines 307-326. Fragmented and disorganized presentation. Lacks integration among the three pathway enzymes.

Does not discuss biological significance of enzymatic sequence. Without correlation with final products (lignin).

The author’s answer:

We appreciate the reviewers' insightful comments! This section has been added to the discussion part, with the following specific additions:  Line 377-386, In the phenylpropanoid pathway, PAL is a crucial enzyme that transforms phenylalanine into trans-cinnamic acid. The production of lignin and phenolics, which are directly linked to fruit disease resistance, is facilitated by three essential enzymes (PAL, C4H, 4CL) in the phenylpropanoid pathway[36,37]. The synergistic interaction of these three components forms the core metabolic pathway: PAL→C4H→4CL. The intermediate product catalyzed by 4CL polymerizes under the action of downstream enzymes to form lignin, enhancing the mechanical strength and degradation resistance of cell walls to form a physical barrier to infection. It also synthesizes secondary metabolites such as total phenolics and flavonoids, which exhibit potent antibacterial activity[36,37]

12、Synergism Mechanism NOT Explained (Lines 327-407)

The author’s answer:

Thank you for highlighting the core issue! The shortcomings in the discussion section regarding the analysis of the synergistic mechanism between water stress and methyl jasmonate (MeJA) were indeed an oversight on our part. The primary focus of this work was to validate that Slight Water Loss shares a similar mechanism of action with MeJA. In our next research phase, we will conduct an in-depth investigation into the specific mechanisms underlying water stress.

13、Superficial Literature Integration (Throughout)

The author’s answer:

We appreciate the reviewers' insightful comments! As our findings are preliminary research results and have not undergone in-depth investigation, we have now revised and consolidated the discussion section of the full text. Moving forward, we will conduct further research based on your suggestions.

14、Lack of Critical Evaluation of Own Results

The author’s answer:

Thank you for highlighting the core issue! Regarding the lack of critical evaluation of our own findings, we acknowledge our shortcomings and the superficial nature of our research. We have addressed this by adding t The specific mechanisms underlying the synergistic effects of water loss and MeJA remain largely unexplored, warranting future research into the detailed mechanisms of this treament combination.(Line 414-416). The Future Research Institute will conduct in-depth research into specific mechanisms.

15、Thank you for highlighting the core issue! Inadequate Discussion of Practical Implications

The author’s answer:

Thank you for highlighting the core issue! Regarding the insufficient discussion of practical applications, we have added the following content to the text: This study demonstrates that potharvest water loss and MeJA application can be used to prevent postharvest diseases in fruits and vegetables. The specific mechanisms underlying the synergistic effects of water loss and MeJA remain largely unexplored, warranting future research into the detailed mechanisms of this treament combination (Line412-416).

16、Why A. arguta responses might differ from other fruits

The author’s answer:

We appreciate the reviewers' insightful comments! Our team primarily studies Actinidia arguta. This research reveals that A. arguta exhibits synergistic induction of stress resistance and phenylpropanoid metabolic responses under water stress + methyl jasmonate (MeJA) treatment. Whether significant differences exist compared to other fruits warrants further investigation.

17、 The discussion fails to identify specific hypotheses or experiments needed to address knowledge gaps.

The author’s answer:

Thank you for pointing out the key issues! We have added the following to the discussion section: This study demonstrates that potharvest water loss and MeJA application can be used to prevent postharvest diseases in fruits and vegetables. The specific mechanisms underlying the synergistic effects of water loss and MeJA remain largely unexplored, warranting future research into the detailed mechanisms of this treament combination.(Line412-416).

18、Conclusions provide no guidance on how findings could be applied commercially or what conditions are required for practical implementation. Conclusions fail to identify key knowledge gaps or propose specific next steps for advancing the field.

The author’s answer:

Thank you for pointing out the key issues! We have revised the conclusion section of the revised manuscript as follows:We confirmed that slight water loss and MeJA share a similar mechanism of action. Furthermore, we found that slight water loss combined with MeJA treatment increased the resistance of A. arguta fruits to gray mold, which may be closely related to the activation of ROS and phenylpropanoid metabolism. This treatment increased the activity of enzymes related to ROS and phenylpropanoid metabolism, increased the content of secondary metabolites, and up-regulated the key enzyme genes for phenylpropanoid metabolism. This combined treatment demonstrates significant efficacy against gray mold disease in Actinidia arguta by inducing fruit disease resistance and enhancing the synthesis of defense-related substances. However, its field application effectiveness and adaptability across different cultivars and storage conditions require further validation.The section on how research findings can be applied to business practices and the conditions required for practical implementation will be added to the discussion section.

Reviewer 2 Report

Comments and Suggestions for Authors

The manuscript "Slight water loss combined with methyl jasmonate treatment improves Actinidia arguta resistance to gray mold by modulating reactive oxygen species and phenylpropane metabolism" is of interest to the scientific community. I offer some considerations and questions below:

- You could discuss also about the treatment (water loss combined with methyl jasmonate) in comparison the other treatment methods of gray mold in Actinidia arguta/kiwifruits during decay reported in literature, such as "hot water", "heat treatment", among other. 

- Please, increase the font size in the graphs of Figures 2 to 5, including those on the graph axes. Furthermore, you could position the legend of color bars on the side of graphs (Figures 3 and 5) and in the empty space (below graph-b, Figures 2 and 4).

- Exclude the first paragraph of discussion:
"Authors should discuss the results and how they can be interpreted from the perspective of previous studies and of the working hypotheses. The findings and their implications should be discussed in the broadest context possible. Future research directions may also be highlighted." (lines 327-329), since it is related to a guideline. 

- Format references in accordance with MDPI Author guidelines (https://www.mdpi.com/journal/information/instructions) 

Suggestions:

- Use "methyl jasmonate" instead of "MeJA" in Keywords;
- Include "phenylpropane metabolism" in Keywords.

Minor corrections:

- Format to italic scientific names "Actinidia arguta" and "Botrytis cinérea" throughout the text and title;
- Format the "6" and "-1" to superscript style in "1 × 106 spores mL-1" (line 96). 

Author Response

1、You could discuss also about the treatment (water loss combined with methyl jasmonate) in comparison the other treatment methods of gray mold in Actinidia arguta/kiwifruits during decay reported in literature, such as "hot water", "heat treatment", among other. 

The author’s answer:

Thank you for your suggestion. Since the current literature on Slightwater loss combined with MeJA treatment focuses on this specific approach, we have only conducted comparisons of individual treatments.

2、Please, increase the font size in the graphs of Figures 2 to 5, including those on the graph axes. Furthermore, you could position the legend of color bars on the side of graphs (Figures 3 and 5) and in the empty space (below graph-b, Figures 2 and 4). 

The author’s answer:

We sincerely appreciate the valuable comments. We have enlarged the font size of the charts in Figures 2 through 5 as per your suggestion, including the text on the axes. Additionally, the color legend has been positioned appropriately within the charts.

3、Exclude the first paragraph of discussion:
"Authors should discuss the results and how they can be interpreted from the perspective of previous studies and of the working hypotheses. The findings and their implications should be discussed in the broadest context possible. Future research directions may also be highlighted." (lines 327-329), since it is related to a guideline. 

The author’s answer:

Thanks for the suggestion. We have removed the first paragraph of the discussion section. We made revisions in the discussion section and added the following:This study demonstrates that potharvest water loss and MeJA application can be used to prevent postharvest diseases in fruits and vegetables. The specific mechanisms underlying the synergistic effects of water loss and MeJA remain largely unexplored, warranting future research into the detailed mechanisms of this treament combination.(Line412-416).

4、Format references in accordance with MDPI Author guidelines

The author’s answer:

Thank you for your valuable comments. We have carefully revised the entire manuscript and formatted the references according to the MDPI author guidelines.

5、- Format to italic scientific names "Actinidia arguta" and "Botrytis cinérea" throughout the text and title;

The author’s answer:

Thanks for the suggestion. We have reviewed the entire text and changed all Latin names of plants and fungi to italicized text.

6、Format the "6" and "-1" to superscript style in "1 × 106 spores mL-1" (line 96)

The author’s answer:

We sincerely appreciate the valuable comments. We have made the corresponding modifications in the text.

Reviewer 3 Report

Comments and Suggestions for Authors

Review of article no.foods-3994906 entitled "Slight water loss combined with methyl jasmonate treatment improves Actinidia arguta resistance to gray mold by modulating reactive oxygen species and phenylpropane metabolism."

The authors investigated an important aspect of increasing kiwiberry fruit's resistance to fungal pathogens during the post-harvest period. Research using methyl jasmonate is not new, but such studies have not been conducted on kiwiberries before. The presented results may be interesting to the reader, but in its current state, the manuscript requires numerous corrections and additions. Unfortunately, the publication was prepared without due diligence. This is evidenced by large sections of text unrelated to the manuscript (remnants of the template), which were not noticed by any of the authors during preparation. All comments are noted in the attached PDF of the manuscript, and the most important ones are presented below, referring to individual chapters.

Introduction:

Please check your citations, as not all are consistent with the content to which they are assigned. For example, items 12 and 13. In item 2, the authors provide data that differs from the data in the cited publication (Line 35). Some statements require corrections (Lines 33 and 39). Latin names of plants and fungi should always be written in italics. Abbreviations of plant names require no further explanation, as their spelling is widely known and defined in the rules of botanical nomenclature.

Material and Methods:

This section requires the most additions, as the description is too terse. I suggest presenting the experiment diagram instead of Table 1, as it adds little to the work, as it largely duplicates the content provided in the description.

The Template sections, unrelated to the manuscript content (Lines 97-107), should be removed.

The degree of ripeness of the tested fruit should be provided (Brix at harvest), and how it was stored after harvest until the experiment. The description of the pathogen preparation requires more detail, for example, the duration of incubation. The chemical analysis descriptions lack information on which fruit samples were used (fresh or frozen). How were such small samples collected for analysis? Were the fruit previously homogenized? In what quantities? Were all replicate fruits homogenized, and samples taken from these samples for each analysis? How did the authors calculate the number of fruits in each treatment? Which parameters were examined for fresh fruit and which for frozen fruit? Was the assessment performed daily during storage? Or after 5 days? Or on days 3, 4, and 5? Which analyses were performed when? This should be clearly stated in the manuscript.

The authors confuse fungi (B. cinerea) with bacteria, which should not be the case in a scientific article (Line 91; the title of section 2.2).

Some methods require a more detailed description or citation of the publication where they are described (Line 175).

Results:

The font size of the descriptions in all graphs is too small, making the graphs unreadable. Below the graph, it should be clarified which results the homogeneous group designations refer to, whether for each term separately or all together. This information should be standardized under each graph.

Discussion:

Template fragments unrelated to the manuscript content should be removed (Lines 327-330). Minor editorial comments in the attached PDF.

Abbreviations:

In my opinion, the explanation of abbreviations should be provided before the manuscript text (unless the publisher indicates otherwise) and, independently, should be explained under the figures. Anyway, botanical names and their abbreviations require no explanation. This is the generally accepted writing convention.

References:

The style of cited publications requires correction and should be consistent with the journal's requirements.

Author Response

1、Please check your citations, as not all are consistent with the content to which they are assigned. For example, items 12 and 13. In item 2, the authors provide data that differs from the data in the cited publication (Line 35). Some statements require corrections (Lines 33 and 39). Latin names of plants and fungi should always be written in italics. Abbreviations of plant names require no further explanation, as their spelling is widely known and defined in the rules of botanical nomenclature.

The author’s answer:

We sincerely appreciate the valuable comments. We have verified that all citations in the introduction section correspond to their assigned content and have formatted all botanical and fungal Latin names in the text using italics.

2、 I suggest presenting the experiment diagram instead of Table 1, as it adds little to the work, as it largely duplicates the content provided in the description.

The author’s answer:

Thank you for your valuable comments. We use Table 1 to make things look simpler and clearer, which is more convenient than text.

3、The Template sections, unrelated to the manuscript content (Lines 97-107), should be removed.

The author’s answer:

Thank you for your valuable comments. This section has been removed.

4、The degree of ripeness of the tested fruit should be provided (Brix at harvest), and how it was stored after harvest until the experiment. The description of the pathogen preparation requires more detail, for example, the duration of incubation. The chemical analysis descriptions lack information on which fruit samples were used (fresh or frozen). How were such small samples collected for analysis? Were the fruit previously homogenized? In what quantities? Were all replicate fruits homogenized, and samples taken from these samples for each analysis? How did the authors calculate the number of fruits in each treatment? Which parameters were examined for fresh fruit and which for frozen fruit? Was the assessment performed daily during storage? Or after 5 days? Or on days 3, 4, and 5? Which analyses were performed when? This should be clearly stated in the manuscript.

The author’s answer:

Thank you for your valuable comments. We have made detailed revisions to this section, as follows:

The A. arguta variety “Longcheng II” was selected, with fruits harvested from Kuandian, Dandong, China, in September 2023. At harvest, the fruits showed a uniform shape and size, with a total soluble solids content of 7 ± 0.2% and no visible damage from pests or diseases on their surface. The harvested A. arguta fruits were immediately transported to the laboratory within 3 h (Line 93-97).

The B. cinerea strain was incubated on potato dextrose agar plates under controlled conditions (26 ± 0.5°C) for 1 wk. The activated pathogenic microorganisms were rinsed with 20 mL of sterile aqueous solution supplemented with 0.05% (v/v) Tween-20. The concentration was further adjusted to ultimately obtain a standardized concentration of 1 × 10⁶ spores mL-1. (Line101-105).

Fifteen fruits were randomly selected from each group for testing. Sequential sampling was conducted on Days 0, 1, 2, 3, 4, and 5 of storage to measure the indicators (disease severity index and lesion diameter) of the fresh fruits. Each replicate fruit pulp segment was homogenized separately, then pooled and homogenized (all replicates were processed individually). Samples were immediately frozen using liquid nitrogen and stored at -80°C for subsequent analysis of total phenolics, lignin content, enzyme activity, and gene expression (Line125-131).

5、The authors confuse fungi (B. cinerea) with bacteria, which should not be the case in a scientific article (Line 91; the title of section 2.2).

The author’s answer:

Thank you for your valuable comments. We sincerely apologize for the confusion between the fungus (B. cinerea) and bacteria; we have now corrected it to B. cinerea.

6、Some methods require a more detailed description or citation of the publication where they are described (Line 175).

The author’s answer:

Thank you for your valuable comments. We have added the cited publications to the revised manuscript and intend to describe the methodology.

7、The font size of the descriptions in all graphs is too small, making the graphs unreadable. Below the graph, it should be clarified which results the homogeneous group designations refer to, whether for each term separately or all together. This information should be standardized under each graph.

The author’s answer:

Thank you for your valuable comments. We have enlarged the font size and other textual elements in the figure within the text. Different lowercase letters in the figure indicate significant differences among treatment groups under the same indicator (p<0.05, Duncan's multiple comparison test). The format for all figure captions remains consistent.

8、Template fragments unrelated to the manuscript content should be removed (Lines 327-330). Minor editorial comments in the attached PDF.

The author’s answer:

Thank you for your valuable comments. This section has been removed.

9、In my opinion, the explanation of abbreviations should be provided before the manuscript text (unless the publisher indicates otherwise) and, independently, should be explained under the figures. Anyway, botanical names and their abbreviations require no explanation. This is the generally accepted writing convention.

The author’s answer:

Thanks for the question. We have removed the plant names and their abbreviations. Regarding the placement of the abbreviation explanations, we will follow the publisher's instructions.

10、The style of cited publications requires correction and should be consistent with the journal's requirements.

The author’s answer:

Thank you for your valuable comments. The citation format has been revised in accordance with the journal's requirements.

Reviewer 4 Report

Comments and Suggestions for Authors

The study investigates the combined effect of slight water loss and methyl jasmonate (MeJA) treatment on gray mold resistance in Actinidia arguta fruit, focusing on defense-related biochemical and molecular responses. The topic fits well within the journal’s scope, addressing postharvest disease management through natural, non-chemical strategies and the underlying resistance mechanisms. The research is timely and potentially impactful; however, several components of the manuscript require clarification and correction to meet scientific and reporting standards. 

Abstract

  • Overstates findings and implies causality not proven by the data (only correlations shown).
  • Claims disease suppression is attributed solely to ROS and phenylpropanoid pathways are too strong.

Introduction

  • Background provided, but research gap and rationale unclear.
  • Needs clearer framing: why this combination and what novelty it adds.
  • Recommend acknowledging potential trade-offs and citing supporting literature.

Materials and Methods

  • Sample size information is contradictory and confusing (replicate number vs. fruit count).
  • Unclear whether replicates are biologically independent.
  • Water-loss procedure lacks detail (consistency among fruits, individual vs. batch weight).
  • Storage conditions, randomization, and cross-contamination prevention not described.
  • Missing non-inoculated control group, this limits interpretation of physiological effects.
  • Biochemical assay descriptions incomplete and sometimes inconsistent.
  • RT-qPCR conditions, reference gene use not clearly explained.
  • Statistical methods unclear and contradictory (LSD vs. Duncan’s).

Results

  • Statistical significance often implied rather than explicitly stated.
  • Excessive descriptive detail without clear synthesis of key findings.
  • No results reported for fruit quality. This contradicts stated aims.
  • Some information misplaced (primers reported in Results instead of Methods).

Discussion

  • Interprets enzyme and phenolic changes as causal without proof, should state correlation only.
  • Lacks critical evaluation of trade-offs and practical applications (e.g., fruit shriveling, MeJA cost/regulation).
  • No consideration of limitations affecting generalizability (single cultivar, ambient storage only).

Conclusion. Recommend a more measured statement: treatment effective under tested conditions and requires further validation.

Author Response

1、Overstates findings and implies causality not proven by the data (only correlations shown).Claims disease suppression is attributed solely to ROS and phenylpropanoid pathways are too strong.

The author’s answer:

We sincerely appreciate the valuable comments. Since we have not conducted extensive research on the mechanism of slight water loss, we have only undertaken preliminary investigations. Therefore, we have revised this section to This effect may be attributed to activation of ROS metabolism, induction of phenylpropanoid metabolism, and up-regulation of related genes, which enhanced the resistance of the fruit to gray mold (Line26-29).

2、Background provided, but research gap and rationale unclear. Needs clearer framing: why this combination and what novelty it adds. Needs clearer framing: why this combination and what novelty it adds.

The author’s answer:

Existing studies have confirmed that MeJA treatment can induce fruit disease resistance, but the synergistic effects of slight water loss and combined stress have not been reported in kiwifruit, and the interaction mechanism between the two remains speculative. As a preliminary exploration, this study aims to validate that slight water loss and MeJA share similar mechanisms. It also demonstrates that the combined treatment effectively controls gray mold disease in Actinidia arguta and provides an initial analysis of the response characteristics of reactive oxygen species and phenylpropanoid metabolic pathways. This lays the groundwork for subsequent in-depth mechanism studies by offering foundational data and directional references. Future research will focus on mechanism elucidation and trade-off optimization (e.g., regulating stress duration and concentration) to further validate the practical application value of combined treatments.

3、Sample size information is contradictory and confusing (replicate number vs. fruit count). Unclear whether replicates are biologically independent.

The author’s answer:

Thank you for your valuable comments. We have made detailed revisions to this section, as follows: Fifteen fruits were randomly selected from each group for testing. Sequential sampling was conducted on Days 0, 1, 2, 3, 4, and 5 of storage to measure the indicators (disease severity index and lesion diameter) of the fresh fruits. Each replicate fruit pulp segment was homogenized separately, then pooled and homogenized (all replicates were processed individually). Samples were immediately frozen using liquid nitrogen and stored at -80°C for subsequent analysis of total phenolics, lignin content, enzyme activity, and gene expression(Line125-131).

4、 Water-loss procedure lacks detail (consistency among fruits, individual vs. batch weight). Storage conditions, randomization, and cross-contamination prevention not described. Missing non-inoculated control group, this limits interpretation of physiological effects.

The author’s answer:

Thanks for the question. Laboratory blowers were used to dry the fruit with cold air, inducing slight dehydration. Storage conditions were maintained at 20±0.5°C with relative humidity (RH) kept between 72-78%. Temperature, humidity, and fruit weight were measured periodically throughout the dehydration process. During dehydration, the fruit positions were rotated at regular intervals while keeping the fruit in bags to prevent cross-contamination, ensuring their relative positions remained unchanged. In subsequent experiments, we will consider comparisons with unvaccinated controls. This study primarily focuses on the effects of vaccination on Actinidia arguta fruit.

5、Biochemical assay descriptions incomplete and sometimes inconsistent.

The author’s answer:

 Thank you for pointing out these issues! We have thoroughly revised the biochemical experimental methods section, adding missing details and standardizing terminology to ensure completeness and consistency.

6、RT-qPCR conditions, reference gene use not clearly explained RT-qPCR

The author’s answer:

Thanks for the suggestion. We have revised this section as follows:Line 172-181,We extracted the total RNA from A. arguta tissues using a cDNA synthesis kit (BioWorks). Analysis using geNorm and NormFinder software ultimately identified Actin as the optimal housekeeping gene. Gene sequences encoding key phenylpropanoid metabolic enzymes in Actinidia arguta—AaPAL, Aa4CL, AaC4H, and AaC3'H—were obtained. Specific primers were selected using the National Center for Biotechnology Information database and synthesized through bioengineering (Table 2). We performed real-time qPCR amplification using the UltraSYBR Mixture (Jiangsu Kangwei Century Biotechnology Co., Ltd., China). Each experiment used three cDNA samples, each of which was prepared from three biological replicates. The comparative 2-ΔΔCT method was employed to compute relative expression levels of target genes[19].

7、Statistical methods unclear and contradictory (LSD vs. Duncan’s)

The author’s answer:

Thanks for the question. We have modified the statistical methods as follows:Line 202-207, All experimental results are based on three independent biological and technical replicates. Each sample underwent at least three parallel analytical determinations. All data are presented as the mean and its corresponding standard deviation. Univariate analysis of variance was performed using SPSS software. Intergroup comparisons were conducted using Duncan's multiple range test, with P < 0.05 as the statistical significance threshold. All graphs were plotted using Origin 2018.

8、Statistical significance often implied rather than explicitly stated.Excessive descriptive detail without clear synthesis of key findings. No results reported for fruit quality. This contradicts stated aims.Some information misplaced (primers reported in Results instead of Methods).

The author’s answer:

Thank you for your suggestion. We have completed all revisions regarding the significance of the conclusions section. The descriptions in the conclusions section have been modified, and a clear summary of key findings has been added. The third objective of this paper has been revised to investigate the effect of combined treatments on the disease resistance of Actinidia arguta fruit. The placement of the primer report will be adjusted according to the editor's suggestions.

9、Interprets enzyme and phenolic changes as causal without proof, should state correlation only.Lacks critical evaluation of trade-offs and practical applications (e.g., fruit shriveling, MeJA cost/regulation).No consideration of limitations affecting generalizability (single cultivar, ambient storage only).

The author’s answer:

Thanks for the suggestion. We have revised the interpretation of enzyme and phenolic changes to n this study, we observed synchronized changes in enzyme activity and phenolic compounds, but the causal relationship between the two and the specific regulatory mechanisms require further validation through experiments such as gene silencing and in vitro enzymatic reactions.(Line 394-398). We strive to ensure the rigor of our conclusions. Our next step will involve studying the combination of water loss and coating films to preserve fruit sensory qualities. MeJA is a natural plant hormone approved for use on fruits and vegetables. Our team primarily studies Actinidia arguta. This research reveals that A. arguta exhibits synergistic induction of stress resistance and phenylpropanoid metabolic responses under water stress + methyl jasmonate (MeJA) treatment. Whether significant differences exist compared to other fruits warrants further investigation.

10、Recommend a more measured statement: treatment effective under tested conditions and requires further validation.

The author’s answer:

Thank you for pointing out the key issues! We have revised the conclusion section of the revised manuscript as follows: We confirmed that slight water loss and MeJA share a similar mechanism of action. Furthermore, we found that slight water loss combined with MeJA treatment increased the resistance of A. arguta fruits to gray mold, which may be closely related to the activation of ROS and phenylpropanoid metabolism. This treatment increased the activity of enzymes related to ROS and phenylpropanoid metabolism, increased the content of secondary metabolites, and up-regulated the key enzyme genes for phenylpropanoid metabolism. This combined treatment demonstrates significant efficacy against gray mold disease in Actinidia arguta by inducing fruit disease resistance and enhancing the synthesis of defense-related substances. However, its field application effectiveness and adaptability across different cultivars and storage conditions require further validation.

Round 2

Reviewer 1 Report

Comments and Suggestions for Authors

Dear Authors,

I value the meticulous adjustments made to address all crucial methodological and interpretive issues highlighted in the review. These corrections show the right scientific rigor: the phenylpropanoid pathway nomenclature has been corrected, disease severity calculations have been clarified to be relative to a quantitative baseline, the enzymatic data has been converted to absolute values (U mg⁻¹ protein), and total phenolic content has been re-measured using the right spectrophotometric method (760 nm). The new mechanistic description of PAL→C4H→4CL  provides the long-missing biochemical basis of this pathway integration.

Although a complete mechanistic elucidation of water stress-MeJA synergism is beyond the scope of this preliminary investigation, your explicit recognition of the knowledge gap and promise of future research is scientifically acceptable. 

< !--a=1-->< !--a=1-->< !--a=1-->< !--a=1-->

Author Response

Dear Editors and Reviewers:

Thank you for your letter and for the reviewers' comments concerning our manuscript entitled "Slight water loss combined with methyl jasmonate treatment improves Actinidia arguta resistance to gray mold by modulating reactive oxygen species and phenylpropanoid metabolism” (lD: foods-3994906) Those comments are all valuable and very helpful for revising and improving our paper, as well as the important guiding significance to our researches. We have studied comments carefully and have made correction which we hope meet with approval. Revised portion are marked with different colors in the paper. The main corrections in the paper and the responds to the reviewer's comments are as flowing:

Reviewer #1:

Comments 1:Does the introduction provide sufficient background and include all relevant references?

Response 1: Thank you for pointing this out. We agree with this comment. Therefore, we have added the species name to the introduction of the revised manuscript as per your suggestion. On lines 37-38, Actinidia arguta (Siebold & Zucc.) Planch. ex Miq., also known as kiwiberry, baby kiwi, or hardy kiwi, is a representative plant of the genus Actinidia. We have further supplemented the requirements for gray mold infection and the current fungicide restrictions in lines 46-49 and lines 53-54. Fungicide restrictions are also described in the discussion section, lines 330-335.

Line46-49: When the fruit peel is damaged during harvesting, transportation, or storage, gray mold fungus (Botrytis cinerea), a typical necrotrophic pathogenic fungus, may adhere to the exocarp and invade the flesh tissue, accelerating fruit decay[5,6]. Wounds and senescent tissues serve as the most critical entry points for invasion.

Line 53-54:  While chemical control remains the primary method for managing gray mold disease in fruit, it readily induces resistance.

Line 330-335:Currently, chemical application is the main method for gray mold control owing to its low cost and rapid effects[20,21]. However, the long-term use of chemicals may induce resistance in pathogenic bacteria. Furthermore, pesticide residues may remain on the surface of fruits, affecting the nutrient composition and sensory quality as well as posing potential food safety risks. Therefore, the development of safe, natural control technologies has gained increasing attention in the prevention and control of gray mold.

Since our investigation into the synergistic effects of slight water loss and MeJA remains preliminary, we have revised the objective of this paper to (1) compare the inhibitory effects of slight water loss and MeJA treatment on gray mold disease in postharvest A. arguta, (2) reval the preliminary disease-resistance induction mechanism of slight water loss combined with MeJA, and (3) investigate the effects of combined treatment on A. arguta postharvest disease resistance. Thank you for suggesting these in-depth directions for the mechanism study. We will conduct further experiments in our subsequent research. I have made the above revisions. Please review this document. If revisions are needed, please specify the paragraphs requiring modification. Thank you for your hard work.

Comments 2:Is the research design appropriate?

Response 2: Thank you very much for your valuable comment on the appropriateness of the research design. We fully agree with your opinion and acknowledge that there were deficiencies in the original research design which may affect the scientific rigor and reliability of the results. After careful discussion and revision, and the specific improvements are as follows: This experimental design builds upon the laboratory's published article “Slight water loss affects the quality of ‘Longcheng 2’ kiwiberry fruit infected with gray mold disease,” further examining the effects of slight water loss, methyl jasmonate, and their combined application on Actinidia arguta fruit inoculated with Botrytis cinerea. Therefore, this experiment did not include a control group inoculated with sterile water for comparison. Future studies will consider incorporating a non-inoculated control group for comparison. The sample size design for experimental materials is based on three biological replicates per treatment group, with three fruits per replicate, meeting the requirements for biological replication. Statistical methods for all experimental results are based on three independent biological and technical replicates. Each sample underwent at least three parallel analytical determinations. All data are presented as the mean and its corresponding standard deviation. Univariate analysis of variance was performed using SPSS software. Intergroup comparisons were conducted using Duncan's multiple range test, with P < 0.05 as the statistical significance threshold. All graphs were plotted using Origin 2018. This method has also been modified according to the experimental design requirements. Please review this document. If revisions are needed, please specify the paragraphs requiring modification. Thank you for your hard work.

Comments 3:Are the methods adequately described?

Response 3: Thank you for your valuable feedback regarding the insufficient description of the methodology. We fully agree with your perspective and have thoroughly revised the “Materials and Methods” section, adding key details to ensure experimental reproducibility. The specific revisions are as follows: Lines 94–97 now include fruit soluble solids content and harvest and transport time. At harvest, the fruits showed a uniform shape and size, with a total soluble solids content of 7 ± 0.2% and no visible damage from pests or diseases on their surface. The harvested A. arguta fruits were immediately transported to the laboratory within 3 h.

The method for preparing the B. cinerea spore suspension has been modified to: The B. cinerea strain was incubated on potato dextrose agar plates under controlled conditions (26 ± 0.5°C) for 1 wk. The activated pathogenic microorganisms were rinsed with 20 mL of sterile aqueous solution supplemented with 0.05% (v/v) Tween-20. The concentration was further adjusted to ultimately obtain a standardized concentration of 1 × 10⁶ spores mL-1 (Lines101-105).

The following has been added to the experimental design and sample collection methods: Fifteen fruits were randomly selected from each group for testing. Sequential sampling was conducted on Days 0, 1, 2, 3, 4, and 5 of storage to measure the indicators (disease severity index and lesion diameter) of the fresh fruits. Each replicate fruit pulp segment was homogenized separately, then pooled and homogenized (all replicates were processed individually). Samples were immediately frozen using liquid nitrogen and stored at -80°C for subsequent analysis of total phenolics, lignin content, enzyme activity, and gene expression (Lines 125-131).

The disease severity index measurement method now includes a classification of rot severity levels. Lines 141-143, disease severity index as follows: Grade 0, no decay observed; Grade 1, decay area 0-25%; Grade 2, decay area 26-50%; and Grade 3, decay area > 50%.

The methods for measuring POD, PPO, and SOD activity have been modified to The POD and PPO activities were determined according to previously reported protocols[14]. Absorbance was measured at wavelengths of 470 nm and 420 nm, respectively, with results expressed as U mg⁻¹ protein.The SOD activity was assessed using an SOD assay kit (ADS-F-KY011; Quanzhou), with results expressed as U mg⁻¹ protein (Lines 146-150).

The method for determining total phenols has been modified to the total phenol content was assayed using a modified version of a previously published method[16]. We homogenized 1.0 g of A. arguta tissue sample in 80% ethanol, added distilled water, NaCO₃, and Folin reagent; and mixed them thoroughly. The supernatant was collected for analysis, and the absorbance was measured at a wavelength of 760 nm (Lines161-165).

The assay method for measuring the expression of the corresponding gene encoding the key enzyme in phenylalanine metabolism was also modified accordingly. Lines 172-181, We extracted the total RNA from A. arguta tissues using a cDNA synthesis kit (BioWorks). Analysis using geNorm and NormFinder software ultimately identified Actin as the optimal housekeeping gene. Gene sequences encoding key phenylpropanoid metabolic enzymes in Actinidia arguta—AaPAL, Aa4CL, AaC4H, and AaC3'H—were obtained. Specific primers were selected using the National Center for Biotechnology Information database and synthesized through bioengineering (Table 2). We performed real-time qPCR amplification using the UltraSYBR Mixture (Jiangsu Kangwei Century Biotechnology Co., Ltd., China). Each experiment used three cDNA samples, each of which was prepared from three biological replicates. The comparative 2-ΔΔCT method was employed to compute relative expression levels of target genes[19].

The statistical analysis section has also been revised accordingly. Lines 202-207, All experimental results are based on three independent biological and technical replicates. Each sample underwent at least three parallel analytical determinations. All data are presented as the mean and its corresponding standard deviation. Univariate analysis of variance was performed using SPSS software. Intergroup comparisons were conducted using Duncan's multiple range test, with P < 0.05 as the statistical significance threshold. All graphs were plotted using Origin 2018.

Please review this document. If revisions are needed, please specify the paragraphs requiring modification. Thank you for your hard work.

Comments 4:Are the results clearly presented?

Response 4: Thank you for your valuable comment pointing out the insufficient clarity of the results presentation. We fully agree with your opinion and have comprehensively revised the results section to improve logical coherence, data specificity, and figure/table standardization. The specific revisions are as follows:

In the results section of the revised manuscript, we have modified the results figures and icons. Each icon now includes the following: Statistical significance was designated as P < 0.05. Different lowercase letters (a–c) indicate significant differences between treatments at P < 0.05.

We have integrated the relationship between the three pathway enzymes and the final product (lignin) based on your suggestions. A detailed description is provided in the Discussion section. Lines 386-392, The synergistic interaction of these three components forms the core metabolic pathway: PAL→C4H→4CL. The intermediate product catalyzed by 4CL polymerizes under the action of downstream enzymes to form lignin, enhancing the mechanical strength and degradation resistance of cell walls to form a physical barrier to infection. It also synthesizes secondary metabolites such as total phenolics and flavonoids, which exhibit potent antibacterial activity[36,37].

We have further incorporated your suggestion by adding descriptions of the magnitude of data variation and its biological significance. Lines 236-249, However, the POD and PPO activities under the combined treatment were significantly higher than those under the other treatments (P < 0.05). The POD activity was 68.78 U mg -1protein higher under Treament 3 than under the control (93.4 U mg -1protein). The PPO activity was 68.74 U mg -1protein higher under Treament 3 than under the control (91.4 U mg -1protein).

The SOD activity exhibited the same trend across all treatment groups as in the control group. It increased from Day 0 to Day 1, peaked on Day 3, and then began to decline (Fig. 2C). The combined treatment group showed significantly higher SOD activity (1.67 U mg⁻¹ protein) than the control group (2.61 U mg⁻¹ protein) (P < 0.05) on Day 5 after inoculation.

Overall, the combined treatment resulted in a significant increase in POD, PPO, and SOD activities. These three enzymes may decompose hydrogen peroxide and participate in lignin synthesis, thereby reducing damage to fruit cells while strengthening the physical barrier of cell walls, thus enhancing disease resistance.

Lines 317-325, The expression of Aa4CL showed similar trends in the control and treatment groups. On Days 1–5 after inoculation, the relative expression levels of Aa4CL in all three treatment groups exceeded that in the control group, with the combined treatment  group showing a 40% higher expression level compared with that in the control group (P < 0.05). The combined treatment significantly increased the relative expression levels of AaPAL, AaC4H, AaC3'H and Aa4CL in the fruit (P < 0.05). The increase in the relative expression levels of AaPAL, AaC4H, and Aa4CL in the combined treament group corresponded to the synchronous increase in the PAL, C4H, and C3'H enzyme activities, as well as to the increase in total phenolic content.

Please review this document. If revisions are needed, please specify the paragraphs requiring modification. Thank you for your hard work.

Comments 5:Are the conclusions supported by the results?

Response 5: Thank you for your critical comment pointing out that the conclusions were not fully supported by the experimental results. We fully agree with your opinion and have carefully revised the conclusions to ensure strict consistency with the data, eliminate unfounded speculations, and clarify the scope of application. The specific revisions are as follows: Lines 397-404, We also found that the cpombination of slight water loss and MeJA treatment significantly outperformed both the control and individual treatments (P < 0.05). This combination reduced fruit rot and enhanced the accumulation of defense-related enzymes associated with ROS and phenylpropanoid metabolism. In this study, we observed synchronized changes in enzyme activity and phenolic compounds, but the causal relationship between the two and the specific regulatory mechanisms require further validation through experiments such as gene silencing and in vitro enzymatic reactions.

Lines 416-422, Overall these results indicate that activation of ROS metabolism and phenylpropanoid pathway-associated enzyme activity during storage effectively controls gray mold disease in A. arguta while preserving high fruit quality. This study demonstrates that potharvest water loss and MeJA application can be used to prevent postharvest diseases in fruits and vegetables. The specific mechanisms underlying the synergistic effects of water loss and MeJA remain largely unexplored, warranting future research into the detailed mechanisms of this treament combination.

Lines 430-435, Combined water loss and MeJA treatment demonstrated significant efficacy against gray mold disease in Actinidia arguta by inducing fruit disease resistance and enhancing the synthesis of defense-related substances. However, its effectiveness in field applications effectiveness and adaptability across different cultivars and storage conditions require further validation.

Please review this document. If revisions are needed, please specify the paragraphs requiring modification. Thank you for your hard work.

Comments 6:Are all figures and tables clear and well-presented?

Response 6: Thank you for pointing this out. We agree with this comment. Therefore, we have made further modifications to all charts in the document based on your suggestions. Please review this document. If revisions are needed, please specify the paragraphs requiring modification. Thank you for your hard work.

Reviewer 3 Report

Comments and Suggestions for Authors

The authors have implemented most of my suggestions, but a few things still require clarification before the article can be accepted:

  1. Fig. 1. The authors write about statistical significance, but the graph lacks any indication of what differs significantly from what.
  2. Figs. 2-5. It remains unclear whether homogeneous groups are distinguished separately for each term or collectively for all terms. This information should be included under every graph, but is provided only partially under Fig. 2.
  3. Not all botanical names are in Italic (L. 101).

Author Response

Dear Editors and Reviewers:

Thank you for your letter and for the reviewers' comments concerning our manuscript entitled "Slight water loss combined with methyl jasmonate treatment improves Actinidia arguta resistance to gray mold by modulating reactive oxygen species and phenylpropanoid metabolism” (lD: foods-3994906) Those comments are all valuable and very helpful for revising and improving our paper, as well as the important guiding significance to our researches. We have studied comments carefully and have made correction which we hope meet with approval. Revised portion are marked with different colors in the paper. The main corrections in the paper and the responds to the reviewer's comments are as flowing:

Reviewer #3:

Comments 1:Fig. 1. The authors write about statistical significance, but the graph lacks any indication of what differs significantly from what.

Response 1:Thank you for pointing this out. We agree with this comment. Therefore, in line 223, we modified Fig.1 to include the significance of differences.

Comments 2:Figs. 2-5. It remains unclear whether homogeneous groups are distinguished separately for each term or collectively for all terms. This information should be included under every graph, but is provided only partially under Fig. 2.

Response 2:Thank you for pointing this out. We agree with this comment. Therefore, except for Fig. 2, we have added supplementary information below all figures.(Line 225-227、277-278、300-302、329-330)

Comments 3:Not all botanical names are in Italic (L. 101).

Response 3:Thank you for pointing this out. We agree with this comment. Therefore, we have modified B. cinerea on line 101 to italics.